# Comparison of Serum Pharmacodynamic Biomarkers in Prednisone-Versus Deflazacort-Treated Duchenne Muscular Dystrophy Boys

**DOI:** 10.3390/jpm10040164

**Published:** 2020-10-12

**Authors:** Shefa Tawalbeh, Alison Samsel, Heather Gordish-Dressman, Yetrib Hathout, Utkarsh J. Dang

**Affiliations:** 1Department of Biomedical Engineering, Binghamton University, Binghamton, NY 13902, USA; stawalb1@binghamton.edu; 2Department of Pharmaceutical Sciences, School of Pharmacy and Pharmaceutical Sciences, Binghamton University, Binghamton, NY 13902, USA; amsamsel@gmail.com; 3Division of Biostatistics, Children’s National Hospital, Washington, DC 20010, USA; HGordish@childrensnational.org; 4Department of Health Outcomes and Administrative Sciences, School of Pharmacy and Pharmaceutical Sciences, Binghamton University, Binghamton, NY 13902, USA

**Keywords:** Duchenne muscular dystrophy, pharmacodynamic biomarkers, prednisone, deflazacort, glucocorticoids, corticosteroids, safety

## Abstract

Prednisone (Pred) and Deflazacort (Dfz) are commonly used glucocorticoids (GCs) for Duchenne muscular dystrophy (DMD) treatment and management. While GCs are known to delay the loss of ambulation and motor abilities, chronic use can result in onerous side effects, e.g., weight gain, growth stunting, loss of bone density, etc. Here, we use the CINRG Duchenne natural history study to gain insight into comparative safety of Pred versus Dfz treatment through GC-responsive pharmacodynamic (PD) biomarkers. Longitudinal trajectories of SOMAscan^®^ protein data obtained on serum of DMD boys aged 4 to 10 (Pred: *n* = 7; Dfz: *n* = 8) were analyzed after accounting for age and time on treatment. Out of the pre-specified biomarkers, seventeen candidate proteins were differentially altered between the two drugs (*p* < 0.05). These include IGFBP-2 and AGER associated with diabetes complications, and MMP-3 associated with extracellular remodeling. As a follow-up, IGFBP-2, MMP-3, and IGF-I were quantified with an ELISA using a larger sample size of DMD biosamples (Dfz: *n* = 17, Pred: *n* = 12; up to 76 sera samples) over a longer treatment duration. MMP-3 and IGFBP-2 validated the SOMAscan^®^ signal, however, IGF-I did not. This study identified GC-responsive biomarkers, some associated with safety, that highlight differential PD response between Dfz and Pred.

## 1. Introduction

Duchenne muscular dystrophy (DMD) is an X-linked recessive disorder affecting the expression of dystrophin protein, an essential protein that maintains muscle fiber integrity and function [1]. The lack of dystrophin expression leads to muscle inflammation at an early stage of the disease, followed by progressive muscle degeneration and wasting [2]. While there are promising gene therapy and exon-skipping treatments, some of which have received conditional approval from the EMA [3] and FDA [4,5,6,7], these are mutation-specific and only restore a partial amount of truncated dystrophin protein [8]. Hence, DMD patients will continue to need combination therapy using the current standard of care: glucocorticoids (GCs). GC use helps reduce muscle inflammation and delay the loss of motor abilities [9], delay loss of independent ambulation, improve pulmonary function, and delay the onset of cardiomyopathy [10,11,12]. However, a GC regimen can have many side effects, such as weight gain, growth stunting, loss of bone density, hirsutism, Cushingoid features, osteoporosis, hypertension, diabetes, behavioral disturbances, and difficulty in sleeping [12,13,14]. Commonly used steroidal drugs for DMD are prednisone (Pred) and deflazacort (Dfz). Prednisone has been used to treat DMD patients in the USA since the seventies [15]. In February 2017, the FDA approved deflazacort to treat DMD patients aged five years and older [16] although this same drug has been widely used in Europe to treat DMD patients for years. Figure 1 shows structures of prednisone and deflazacort along with their respective active metabolites: prednisolone and 21-desacetyl deflazacort, respectively. Both drugs are given to patients in their prodrug forms, which are then metabolized to their active forms in the liver. The active metabolite prednisolone has a hydroxyl group at carbon 17, while the active 21-desacertyl deflazacort has an oxazoline structure at that same position (red arrow in Figure 1).

Many efforts have been undertaken to investigate the use of biomarkers in DMD [17,18,19] and the efficacy and safety of Pred versus Dfz [10,11,12,20,21]. Both drugs are known to be efficacious in prolonging ambulation [10,11,14,22]. For our purposes here, we provide a small review on safety comparisons from the literature. These studies were compared to clinical outcome measures like changes in height, weight, Cushingoid features, and erythema, among others to evaluate differences in safety. A comparison of high dose Dfz and Pred in 70 systemic lupus patients concluded that Pred was associated with a significant increase in weight gain, Cushingoid severity index, and hirsutism compared to Dfz [23]. In a double-blind, randomized study on 196 DMD subjects, Pred was found to be associated with more weight gain than Dfz [10]. In another randomized study (18 DMD patients), patients treated with Pred showed higher weight gain compared to Dfz-treated patients [13]. As compared to other studies [10,14], a comparison of prednisone/prednisolone versus Dfz treated patients using 340 participants from the Cooperative International Neuromuscular Research Group Duchenne Natural History Study (CINRG-DNHS) showed that participants treated with Dfz had increased frequency of Cushingoid appearance, cataracts, and growth delay. Another study [13] reported that the time of initiating, dosing, and duration of treatments were associated with side effects; longer duration and increased Dfz dosage predicted growth stunting and Dfz was reported to be associated with lighter weight and shorter heights compared to Pred [13]. To summarize, these studies suggest that treatment with Pred is associated with relatively more weight gain, whereas Dfz treatment is associated with relatively more growth stunting. While these studies provide important comparisons between Dfz and Pred, the underlying molecular mechanism leading to these differences, remain unknown. Furthermore, predictive outcome measures, especially for adverse effects are highly desirable.

There are ongoing debates among families, clinicians, and regulatory agencies over which GC drug (Pred or Dfz) and regimen is better for DMD boys. GC responsive biomarkers (for both safety and efficacy) have been defined and confirmed in different diseases and multiple cohorts [17,24,25], using data with pooled corticosteroid drugs not differentiating between Pred and Dfz and different regimens [22]. However, much remains unknown about the comparative effects of Pred vs. Dfz on blood accessible biomarkers and how they can inform clinical decision-making and drug development. Adding pharmacodynamic (PD) biomarkers to the evidence from the aforementioned clinical outcome studies could bring insights into differences between Pred and Dfz at the molecular level. To the best of our knowledge, we are the first to compare serum accessible PD biomarkers in DMD patients (4–16 years old) treated with Dfz or Pred to gain insight into differences at the serum protein level. For this, we use data generated by a high-throughput SOMAScan^®^ technique, followed by confirmation of a specific set of PD biomarkers by ELISAs. The objective here is to define PD biomarkers that differ in their response to Pred and Dfz, and investigate if these biomarkers are known to be associated with reported differences in term of side effects between these two drugs.

## 2. Materials and Methods

The study protocol was approved by Institutional Review Boards at all participating institutions that provided serum samples. These included, the Office of Research IRB administrations at the University of California Davis, Davis, CA, the University of Pittsburgh, Pittsburgh, PA, Children’s National Health System, Washington DC, the Conjoint Health Research Ethics Board, University of Calgary and the Human Subject Research Review Committee at Binghamton University, NY. Informed written consent was obtained from the parents of the participants or their legal guardians for biomarker studies at each site.

### 2.1. SOMAscan^®^ Dataset

The CINRG-DNHS is a natural history study of 440 DMD boys. The CINRG-DNHS investigators enrolled these subjects at 20 centers in nine countries [11]. In this prospective cohort study, subjects genetically confirmed to have DMD were followed for up to 10 years. Details about the study, including informed consent and others, have previously been published [26]. At entry, some DMD subjects were steroid-naïve and were then treated during follow up visits, some remained steroid-naïve throughout, while others were already on GCs [26]. Characteristics (height, weight, age, time on GC treatment, and type of GC) were recorded on follow-up visits as well as clinical outcome data on time to run/walk, time to stand, and time to climb until loss of ability [26]. Note that six-minute walk distance and the NorthStar Ambulatory Assessment score were measured for a small subset of patients enrolled later during the study [26]. Blood samples for biomarker investigations were collected by the CINRG-DNHS investigators during some visits on a subset of participants. GC-treated patients were on Pred or Dfz with variation in dose and dose schedules (intermittent or daily). In a recent study [17], we generated biomarker data on a subset of these DMD samples (*n* = 31 boys, 4 to 10 years old) and used steroid-naïve and steroid-treated paired samples to define 107 (false discovery rate-adjusted *p*-values < 0.05) PD biomarkers from 1310 serum proteins (SOMAscan^®^ array).

Based on the above-mentioned study [17], we pre-specified the 107 PD biomarkers of interest and focused on an analysis of 35 longitudinal samples from 8 Pred-treated and 7 Dfz-treated ambulatory patients only (screening dataset A) to identify potential candidate serum proteins differentially altered between Pred and Dfz. These subjects were age-matched (see Table 1 for a summary of their characteristics). For these longitudinal data, only subjects with at least 2 visits were used. The name of the study, type of treatment, duration of treatment, age, and some clinical measurements (height, weight, BMI) are tabulated in Appendix A.

### 2.2. Validation of Key Pharmacodynamic Biomarkers Using ELISA

A larger data set (confirmation dataset B) contained up to 76 longitudinal samples on 29 ambulatory subjects (17 deflazacort-treated and 12 prednisone-treated) from CINRG DNHS. Note that all samples/subjects from screening dataset A were used in confirmation dataset B as well. For both datasets, we focused on ambulatory patients to minimize any possible confounding issues with the stage of disease (after loss of ambulation). Again, these subjects were age-matched. Summary characteristics of the subjects are provided in Table 2.

A subset of PD serum protein biomarkers, including MMP3, IGFBP-2, and IGF-I, were selected for confirmation analysis using ELISA assays. An ELISA kit from Meso Scale Diagnostics (MSD) was used to measure levels of MMP3. Similarly, ELISA kits from Thermo Fisher Scientific and R&D Systems, Inc. were used to measure levels of IGFBP2 and IGF-I, respectively. These candidates were chosen based on the availability of a validated ELISA assay kit. All three assays were sandwich ELISA, which were highly specific for antigen detection. Dilution factors for serum samples were 1:10, 1:400, and 1:100 for MMP-3, IGFBP-2, and IGF-I, respectively. The three assays were performed following the manufacturer’s instructions. Details about serum samples used in ELISA validation are shown in Table 2.

### 2.3. Data Analysis and Statistical Methods

As depicted in the schematic (Figure 2), we first pre-specified 107 steroidal PD biomarkers previously identified as being significantly altered over time in their levels between GC-naive and paired GC-treated DMD samples from the same subjects [17]. To compare the serum levels and trajectories of these pre-specified 107 PD biomarkers between DMD patients treated with Dfz and Pred, linear mixed effect models were used to analyze longitudinal measurements for 7 boys on Pred and 8 on Dfz (daily regimen). All statistical analyses were performed using R [27]. Linear mixed effect models were run using the lme4 and lmerTest packages [28,29]. Serum protein levels, which had previously been hybridization control and median signal normalized, were log-transformed (see [17] for details). Random intercept linear mixed effect models were used to investigate the association of (mean centered) time on drug, type of GC (Pred or Dfz), and interaction between time on drug and type of GC on log transformed protein RFUs. The samples were age-matched at baseline (average age of Pred-treated subject at baseline = 5.7 (min age = 4.3, max age = 7.3) years; average age of Dfz-treated subject at baseline = 6 (min age = 4.7, max age = 7.4) years) and over time (see Table 1 for age range at sample collection). Some models did not converge (7 out of 107) due to numerical optimization issues (likely due to small sample size). Here, the interaction coefficient represents the difference in estimated slopes of biomarker trajectory over time between Pred-treated and Dfz-treated DMD patients. If the interaction coefficient is significant, this indicates that treatment type (Pred or Dfz) is associated with different trajectories of biomarker over time. The coefficient for the type of GC represents the difference in mean protein levels between Pred-treated and Dfz-treated subjects for a patient with an average treatment duration with similar biomarker trajectories over time (this is considered after the interaction effect). The same model was also fit to log-transformed ELISA data. To study the effect of time on GC on weight, height, and BMI, data analyses were performed using linear mixed effect models on confirmation dataset B. For this, we used a random intercept linear mixed model adjusting for age at baseline, time on GC, and interaction between type of GC and time on GC. Concordance of SOMAscan^®^ and ELISA measurements was investigated on the subset of samples common to screening dataset A and confirmation dataset B.

## 3. Results

### 3.1. Longitudinal Trajectory of Serum PD Biomarkers in Prednisone vs. Deflazacort Treated DMD boys

We examined differences in the longitudinal trajectories of pre-specified 107 PD biomarkers in Pred-treated patients (*n* = 7) versus Dfz-treated DMD patients (*n* = 8) accounting for duration of treatment (time on GC) and interaction between treatment duration and type of GC (Pred or Dfz). In general, Pred and Dfz seem to engage the PD biomarkers similarly and altered their trajectory in the same direction in screening dataset A. This was not the case for 17 PD biomarkers that were found differentially altered in their average levels and/or longitudinal trajectories (unadjusted *p* value < 0.05) in Pred-treated vs. Dfz-treated DMD patients. These 17 differentially altered PD biomarkers are listed in Table 3 with fold change between non-GC-treated DMD subjects and healthy controls (from [17]), fold change between Pred and Dfz-treated subjects, p values for the difference in the mean levels and p values for the difference in longitudinal trajectories between Pred and Dfz treated DMD patients, along with potential significance and references to published literature.

Two major groups of differentially affected steroidal PD biomarkers were thus identified. The first major group consisted of PD biomarkers that were repressed by GC in general, but exhibited a significantly lower mean level in Pred treated group relative to Dfz treated group. These included leukocyte immunoglobulin-like receptor subfamily B member 1 (LILRB1), tumor necrosis factor receptor superfamily member 21 (TNFRSF21), chordin-like protein 1 (CHRDL1), soluble advanced glycosylation end product-specific receptor (sRAGE), annexin A2 (ANXA2), CD166 antigen (CD166), scavenger receptor cysteine-rich type 1 protein M130 (sCD163), induced myeloid leukemia cell differentiation protein (MCL-1), transmembrane glycoprotein NMB (GPNMB), mitogen-activated protein kinase 14 (MAPK14), and neural cell adhesion molecule L1 (NCAM-L1). The second (minor) group consisted of PD biomarkers that increased in their levels following GC treatment and consisted of two subgroups. The first subgroup consisted of those that were significantly elevated in serum samples of Dfz relative to the Pred-treated group such as hemojuvelin (HJV), stromelysin-1 (MMP3) while the second group consists of those biomarkers that were significantly elevated in serum samples of Pred relative to Dfz-treated group such as insulin-like growth factor I (IGF-I), ficolin-1 (FCN1), and cGMP-inhibited 3′,5′-cyclic phosphodiesterase A (PDE3A). One interesting PD biomarker is insulin-like growth factor-binding protein 2 (IGFBP-2); this showed diverging longitudinal trajectories (*p* = 0.040) in the Dfz-treated group, as compared to the Pred-treated group. Similarly, longitudinal trajectories of Ficolin-1 (FCN1; *p* = 0.03) and Annexin A2 (ANXA2; 0.054) also seemingly diverged in addition to their differential levels between the two drugs. Figure 3 shows selected examples of these differentially affected steroidal PD biomarkers. For example, IGFBP-2 sharply decreased over time in Dfz-treated DMD patients, while it remained unchanged or slightly increased over time in Pred treated group (*p* = 0.040). While TNFRSF21 (also known as DR6) was not found to be differentially affected by the two different drugs in terms of longitudinal trajectory slopes but its mean sera levels was significantly lower in the Pred-treated group compared to the Dfz-treated group (*p* = 0.005). Similarly, mean levels of FCN-1 were lower in the Dfz-treated group relative to the Pred-treated group (*p* = 0.038), while MMP3 mean levels were significantly elevated in the Dfz-treated group relative to the Pred-treated group.

### 3.2. Effect of Dfz and Pred on Height and Weight of DMD Boys

We also investigated differences in growth stunting in Dfz treated patients relative to Pred treated patients in confirmation dataset B. DMD patients treated with Dfz had lower height growth rates (*p* = 0.006 for difference in trajectory slopes over time) compared to those treated with Pred (Figure 4). We did not find any significant difference in weight (*p* = 0.112) and BMI (*p* = 0.08) over time between patients treated with Dfz and Pred (Figure 4). Additionally, note that more variation is observed in longitudinal BMI measurements around the estimated model (Figure 4).

### 3.3. Data Validation Using ELISA

For the reported data above, we had a small sample size and we did not adjust for multiple testing, thus a follow-up confirmation analyses on a subset of PD biomarkers was carried out. Unfortunately, from the list of PD biomarkers identified in Table 3 above, a good validated ELISA assay that used low sera volume was available for only three PD biomarker candidates: MMP3, IGFBP2 and IGF-I. There were other ELISA assays for other candidates, but they were either not validated or required a larger volume of sera samples, which we did not have. To confirm the SomaScan^®^ findings obtained for MMP3, IGFBP2, and IGF-1, we used a larger sample size of samples over a longer treatment duration and a wider age range of subjects. The findings from the ELISA runs are summarized in Table 4. Results for MMP-3 validated the SOMAscan^®^ signal (same directionality; p-value for the difference in mean levels = 0.022), IGFBP-2 neared significance (p-value for difference in trajectory slopes = 0.051), while IGF-I did not validate the SOMAscan^®^ signal. Figure 5 shows correlations between SOMAscan^®^ data and ELISA data with the IGF-I measurements having a substantially worse Pearson correlation than MMP-3 and IGFBP-2.

## 4. Discussion

GCs are and continue to be the standard of care for several chronic inflammatory and auto-immune diseases including DMD [11,21,25]. In this study, and due to the interest and ongoing debates between families and clinicians regarding which GC drug is more beneficial and safe for DMD boys, we focused on a subset of Dfz- an Pred-treated DMD patients to define the differential pharmacodynamic response to these two commonly prescribed drugs using blood circulating proteins. We previously defined a set of PD biomarkers that are responsive to GC treatment in DMD [17]. In that previous study, we showed that GC affected the levels of 107 circulating serum proteins. In general, use of GCs decreased the levels of several circulating pro-inflammatory and immune response associated proteins, but caused an increase in certain proteins associated with metabolism and extracellular remodeling [17]. In this current study, we find that, in general, both Dfz and Pred engaged the PD biomarkers in a similar fashion, however, among the 107 PD biomarkers, 17 exhibited a differential longitudinal response to Dfz relative to Pred, in directionality, mean levels, or both.

A close examination of the 17 differentially altered PD biomarkers (from screening dataset A) between Dfz and Pred in term of their levels, longitudinal directionality and their potential physiological function led to the following interpretation. Among these 17 PD biomarkers, 12 were significantly reduced in the Pred-treated group relative to the Dfz-treated group after adjusting for treatment duration (the samples were age matched). These 12 decreased PD biomarkers by Pred can be classified into subgroups with the first subgroup consisting of inflammatory and immune associated proteins such as LILRB1, TNFRSF21, sRAGE, CD166, and sCD163. Previous studies have suggested that Dfz is a stronger immune suppressant than Pred [43]. In contrast, our analysis showed that for the above-mentioned five serum proteins, Pred seemed to be more immunosuppressive than Dfz. However, further studies using a larger sample size and additional cellular biomarkers are needed to claim this finding. Another subgroup that was differentially decreased by Pred relative to Dfz included bone mineralization protein GPNMB and the cell adhesion protein NCAM-L1. GPNMB, also known as osteoactvin in rats, has been shown to be implicated in bone mineralization and bone regeneration [39,40] and the larger decrease in its mean level in the Pred treated group relative to the Dfz treated group could suggest that Pred treated patients might have a higher risk of losing bone density than Dfz treated patients. This agrees with earlier studies showing that decreases in bone density was markedly spared by Dfz, as compared to Pred in both adult and children cohorts [44]. However, further studies correlating low levels of circulating GPNMB to loss of bone density in Pred treated group relative to Dfz treated group are needed to support this hypothesis. The differential decrease in the circulating levels of NCAM-L1 by Pred could be indicative of another potential side effect of Pred over Dfz. Indeed, a recent study linked low levels of circulating NCAM-L1 to risk of developing type 2 diabetes [41], which is in agreement with an earlier study showing that Pred was more diabetogenic than Dfz in a pediatric population [45]. The differential decrease of sRAGE by Pred over Dfz could also be linked to the diabetogenic effect of Pred since sRAGE has been suggested to act as a decoy that dampens the advanced glycation end product signaling (e.g., RAGE signaling) and thus increases the risk of developing diabetes [33].

However, the differential decrease in the mean levels of MCL-1, MAPK14, and CHRDL1 by Pred relative to Dfz could be considered to be evidence of possible efficacy in DMD. Indeed, MCL-1 and MAPK14 were previously reported by us [17] to be elevated in blood of untreated DMD boys relative to age matched healthy controls then decreased by GC treatment toward the levels in healthy controls. CHRDL1, however, is known to bind to BMP4 and antagonizes its function [30]. BMP4 is a member of the member of the transforming growth factor-β (TGF-β), a well-known pathway reported to be involved in DMD pathogenesis [46]. Thus, a differential decrease of CRDL1 by Pred relative to Dfz could suggest a beneficial effect but further experiments using a CRDL1 knockout animal model are needed to verify this hypothesis.

Another relevant PD biomarker that was differentially altered over time between the two drugs is IGFBP-2. It sharply decreased in its longitudinal trajectory in Dfz-treated group compared to the Pred-treated group. This observation was further validated by ELISA assay (*p* = 0.0507; near significance). Note that, in our ELISA confirmations, we had the smallest number of additional samples available for IGFBP-2. IGFBP-2 binds to IGF-1 and regulates growth. While IGFBP-2 was decreased by Dfz treatment, IGF-1 was relatively increased by Dfz over Pred. Unfortunately, we were unable to validate the IGF-1 SomaScan^®^ signal using ELISA; nevertheless, an increase in IGF-1 by GC treatment might be associated with antiinflammation efficacy [31]. However, the selective longitudinal decrease of IGFBP2 by Dfz could be associated with significant growth stunting by Dfz over the Pred reported by others [20,47,48] and confirmed by us in this study. Furthermore, a gene knockout study, conducted on mouse model comparing Igfbp2-/- mice colony with Igfbp2+/+ mice colony determined the role of IGFBP2 in bone turnover and showed that Igfbp2-/- males had shorter femurs and were heavier than controls but were not insulin resistant [42]. The decrease in IGFBP2 could have dual side effects on both growth stunting and risk of bone fracture. Indeed, a recent study compared a cohort of DMD boys treated with Dfz to a cohort treated with hydrocortisone and concluded that DMD boys receiving daily dose of Dfz had a higher incidence of bone fractures with greater risk of growth stunting [47].

The second group of steroidal PD biomarkers that were differentially affected by the two drugs are those that increased following treatment with GC. FCN1, IGF-I and PDE3A were relatively increased by Pred over Dfz, while MMP3 and HJV were relatively increased by Dfz over Pred. While FCN1 mean levels were higher in the Pred treated group relative to Dfz treated group, it decreased over time with Pred treatment while remaining relatively unchanged over time following Dfz treatment. This is intriguing and requires further validation using orthogonal methods such as ELISA. Unfortunately, the available ELISA assay required the use of a larger sample volume, which was a limitation of our study.

A comparison of the relative effects of Pred and dexamethasone (Dex), another glucocorticoid, on short-term growth and bone turnover confirmed that both drugs affected short-term bone turnover and growth [20]. However, Dex may be more potent in suppressing linear growth, simulating weight gain and bone turnover compared to prednisone. Dex was more potent at depressing IGF-I levels than prednisolone. This is consistent with our SOMAscan^®^ data, IGF-1 was significantly more depressed or decreased in Dfz treated group relative to Pred treated group. However, we were not able to confirm this finding using ELISA. The discrepancy between the SOMAscan^®^ data and ELISA data could be associated with an epitope effect. The two techniques recognize different epitopes on the IGF-I. Furthermore, IGF-I might co-exist as free form and bound form to circulating insulin growth factor binding proteins and these might further interfere with both these affinity-based assays that relies on epitope binding. Future analyses using denatured condition and targeted mass spectrometry analysis are needed to examine the true levels of circulating IGF-I.

MMP3, also known as stromelysin-1, was dramatically increased in the blood circulation following GC treatment, as previously shown by us [17] and others in both DMD patients [24] and inflammatory bowel disease patients [25] and anti-neutrophil cytoplasmic antibody-associated vasculitis, and juvenile dermatomyositis [25]. In this study, we further show that this increase is associated more with Dfz use than Pred use. This was further confirmed by an ELISA assay on a confirmation dataset using a larger sample size with longer treatment duration. The mechanism by which GC induces levels of circulating MMP3 remains unclear. Further studies using cell culture expressing MMP3 are needed to explain the difference of action of Dfz and Pred on the expression of MMP3. Owing to the function of MMP3, i.e., degradation of extracellular matrix, this differential increase might result in more extracellular matrix remodeling in Dfz-treated DMD patients relative to Pred-treated patients. Whether this extracellular matrix remodeling is adverse or beneficial remain to be carefully examined.

## 5. Conclusions

Here, we investigated less invasive and objective PD biomarkers that might prove useful to monitor disease progression and response to GC therapies in DMD. We identified differences in serum levels of PD biomarkers between Pred- and Dfz-treated subjects, some of which may be associated with safety. Such blood accessible biomarkers in DMD could play an important role in clinical trials and decision making [49]. In our comparisons, we adjusted for duration of treatment, the comparison groups were age matched, and for a subset of proteins, we conducted ELISA validation testing of SOMAscan^®^ signal. IGFB2 had a decreasing longitudinal trend associated with Dfz, at odds with the observation for Pred. This may be associated with differential growth stunting seen between the two drugs. The dramatic increase of MMP3 by Dfz relative to Pred remains to be interpreted. Further studies are needed to test the physiological significance of these Dfz and Pred differentially affected PD biomarkers. The study’s limitations include the small sample size, no adjustments for multiple testing, and that the data come from a natural history study with lots of observed variability. We have tried to overcome these limitations by performing lab validation using ELISA and using a larger sample size and longer treatment duration. For biological validation, an external cohort is needed to validate our findings. Note also that we did not investigate direct associations of biomarkers with efficacy/safety outcomes; larger sample sizes are required for such studies and are the objective of ongoing research. While preliminary, this study identified serum proteins that were altered between the Dfz and Pred groups using SOMAscan^®^ array data and we successfully validated two biomarkers using ELISA that may be associated with adverse effects. However, further studies using larger sample sizes collected from well controlled cohorts enrolled in ongoing and future clinical trials comparing the safety and efficacy of Dfz versus Pred are needed to validate and test these differentially altered pharmacodynamic biomarkers identified herein.

## Figures and Tables

**Figure 1 jpm-10-00164-f001:**
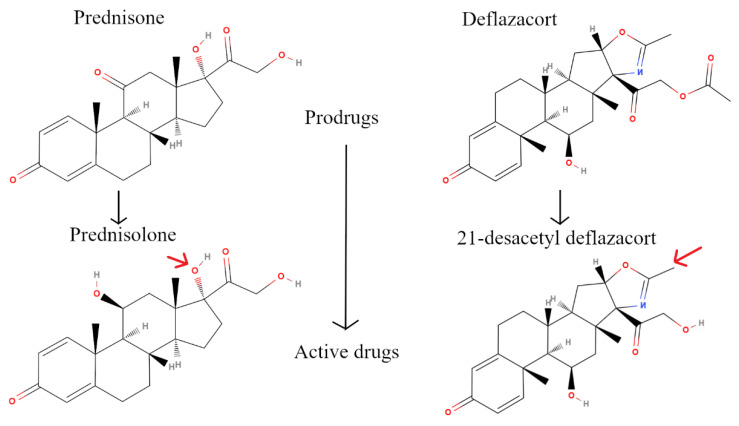
2D structures of Prednisone and Deflazacort (top panels) and their active drugs: prednisolone and 21-desacetyl deflazacort. The similarities are clear. The red arrows point to the structural differences between the active drugs.

**Figure 2 jpm-10-00164-f002:**
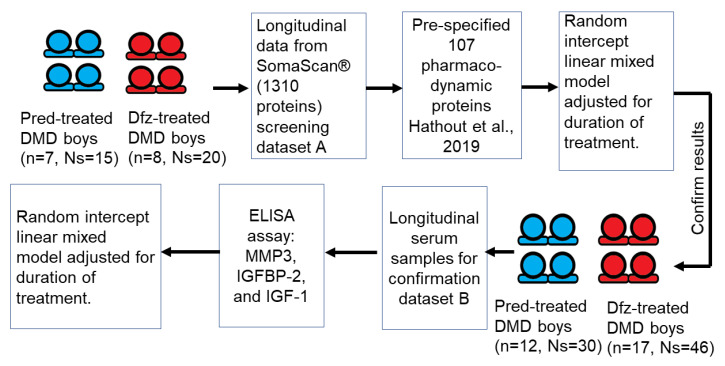
Schematic describing the workflow for the biomarker analyses. Proteins identified as differentially affected by prednisone vs. deflazacort were identified from the SomaScan^®^ based screening dataset. This was followed by confirmation analysis using ELISA assays of three selected biomarkers on a larger set of subjects (confirmation dataset). n: number of subjects, Ns: number of serum samples used in this study.

**Figure 3 jpm-10-00164-f003:**
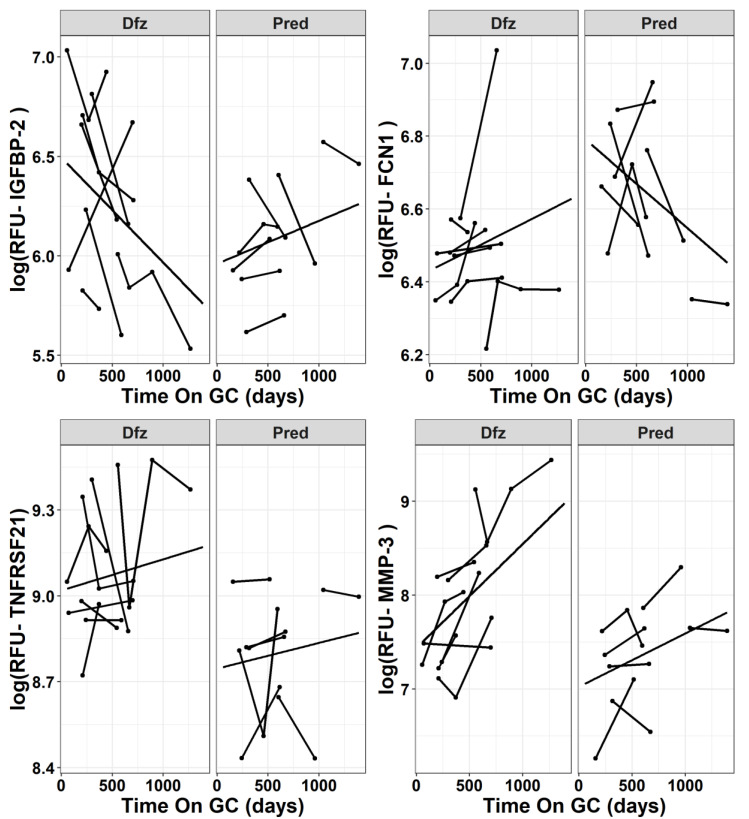
Selected examples of serum protein PD biomarkers showing difference between prednisone (Pred) and deflazacort (Dfz)-treated samples. FCN-1 has relatively higher mean RFU levels in Dfz-treated patients (*p* = 0.038 for mean levels; *p* = 0.03 for difference in trajectory slopes over time). TNFRSF21 has higher mean RFU levels in Pred- vs. Dfz-treated DMD patients (*p* = 0.005 mean levels; *p* = 0.934 for difference in trajectory slopes over time). The longitudinal trajectories of IGFBP-2 levels are different between the two drugs (*p* = 0.04). MMP-3 mean RFU level is elevated in DMD boys treated with Dfz, as compared to Pred (*p* = 0.008).

**Figure 4 jpm-10-00164-f004:**
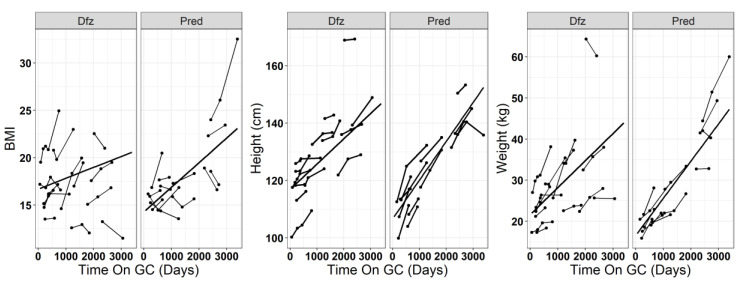
Comparison of longitudinal trajectories between DMD patients treated with deflazacort and prednisone of BMI (*p* = 0.08 for difference in trajectory slopes), height (cm; *p* = 0.006), and weight (kg; *p* = 0.112).

**Figure 5 jpm-10-00164-f005:**
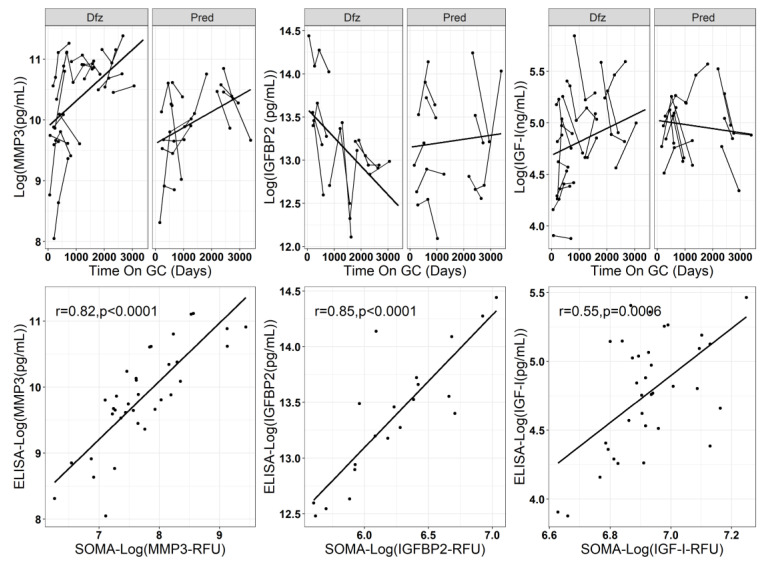
SOMAscan^®^ signal confirmation using ELISA. Upper panel shows longitudinal trajectories of selected DMD serum protein PD biomarkers from ELISA assays. Lower panel shows correlation plots between SOMA and ELISA data for the samples that overlapped between the screening and confirmation data sets.

**Table 1 jpm-10-00164-t001:** Characteristics of patients/samples for SOMAscan^®^ screening dataset A. All samples were from ambulatory patients.

Treatment	Number of Patients	Average Number of Visits (Min, Max)	Age Range at Sample Collection (Years)	Average Time between Biosample Collection (Days) (Mix, Max)	Regimen
**Deflazacort**	8	2.5 (2, 4)	4.7–9.4	465 (56, 1268)	Daily on Dfz
**Prednisone**	7	2.14 (2, 3)	4.3–8.3	582 (157, 1392)	Daily on Pred

**Table 2 jpm-10-00164-t002:** Characteristics of patients/samples for confirmation dataset B. These serum samples were used for ELISA confirmation of MMP3, IGFBP-2 and IGF-1. All the samples were from ambulatory patients.

Treatment	Number of Patients	Average Number of Visits (Min, Max)	Mean Age at Sample Collection in Years (Min-Max)	Average Time between Biosample Collection (Days) (Mix, Max)
Deflazacort	17	2.7 (2, 5)	9 (4.7–15.3)	1392 (370, 3058)
Prednisone	12	2.5 (2, 3)	8.5 (4.2–15.8)	1685 (594, 3391)

**Table 3 jpm-10-00164-t003:** List of PD biomarkers that were differentially altered between Pred-treated (*n* = 7) and Dfz-treated (*n* = 8) groups (SOMAscan^®^ screening dataset A).

Abbreviated Gene Name ^1^ (Uniprot ID)	Fold Change between Untreated DMD vs. Healthy Controls with *p*-Value from [17]	Fold-Change between DMD Subjects Treated with Pred and Dfz	*p*-Value: Difference in Mean Levels ^2^	*p*-Value: Difference in Trajectories ^2^	Protein Function—Biological Process	Potential Significance
LILRB1 (Q8NHL6)	0.96 (0.861)	−1.5 (↓) in Pred vs. Dfz	0.005	0.426	Immune response	Side effect (immune suppression)
TNFRSF21 (O75509)	1.14 (0.454)	−1.33 (↓) in Pred vs. Dfz	0.005	0.934	Apoptotic process, adaptive immune response	Side effect (immune suppression)
CHRDL1 (Q9BU40)	1.30 (0.039)	−1.2 (↓) in Pred vs. Dfz	0.007	0.455	Bone Morphogenetic Proteins (BMP) signaling pathway	Efficacy via action on TGF-β signaling [30]
IGF-I (P05019)	0.85 (0.161)	1.14 (↑) in Pred vs. Dfz	0.007	0.096	Promotes growth	Potential efficacy associated with anti-inflammatory propriety [31]
MMP-3 (P08254)	0.77 (0.488)	2 (↑) in Dfz vs. Pred	0.008	0.216	Extracellular matrix degradation	Efficacy/Side effect [17,25,32]
sRAGE/AGER (Q15109)	0.60 (0.005)	−1.82 (↓) in Pred vs. Dfz	0.010	0.640	Inflammatory (causes complications in diabetes)	Side effect may be associated with diabetes risk [33]
ANXA2 (P07355)	1.24 (0.246)	−1.22 (↓) in Pred vs. Dfz	0.011	0.054	Angiogenesis, biomineral tissue development, inflammation	Potential efficacy marker associated with inflammation [34,35,36]
CD166 (Q13740)	0.81 (0.088)	−1.25 (↓) in Pred vs. Dfz	0.014	0.616	Cell adhesion, adaptive immune response	Side effect (immune suppression)
HJV (Q6ZVN8)	0.83 (0.073)	1.19 (↑) in Dfz vs. Pred	0.017	0.385	BMP signaling pathway, iron ion homeostasis	Unknown
sCD163 (Q86VB7)	0.86 (0.466)	−1.30 (↓) in Pred vs. Dfz	0.025	0.340	Inflammation	Efficacy [37]
Mcl-1 (Q07820)	2.31 (<0.001)	−1.33 (↓) in Pred vs. Dfz	0.029	0.146	Apoptosis, DifferentiationInflammation	Efficacy [38]
PDE3A (Q14432)	1.03 (0.805)	1.22 (↑) in Pred vs. Dfz	0.033	0.956	Cell to cell signaling	Unknown
GPNMB (Q14956)	0.79 (0.111)	−1.32 (↓) in Pred vs. Dfz	0.033	0.944	Cell adhesion, bone mineralization.	Side effect related to bone [39,40]
FCN1 (O00602)	1.02 (0.905)	1.18 (↑) in Pred vs. Dfz	0.038	0.030	Innate immune response	Unknown
MAPK14 (Q16539)	2.10 (<0.001)	−1.18 (↓) in Pred vs. Dfz	0.041	0.982	Potential muscle injury biomarker	Efficacy, muscle injury protein that normalized after GC treatment [17]
NCAM-L1 (P32004)	0.76 (0.072)	−1.28 (↓) in Pred vs. Dfz	0.042	0.691	Cell adhesion and differentiation	Side effect, risk of developing diabetes [41]
IGFBP-2 (P18065)	2.55 (<0.001)	−1.16 (↓) in Dfz vs. Pred	0.329	0.040	Growth regulation	Side effect, growth stunting [42]

^1^ LILRB1 = Leukocyte immunoglobulin-like receptor subfamily B member 1; TNFRSF21 = Tumor necrosis factor receptor superfamily member 21; CHRDL1 = Chordin-like protein 1; IGF-I = Insulin-like growth factor I; MMP-3 = Stromelysin-1; sRAGE/AGER = Advanced glycosylation end product-specific receptor, soluble; ANXA2 = Annexin A2; CD166 = CD166 antigen; HJV = Hemojuvelin; sCD163 = Scavenger receptor cysteine-rich type 1 protein M130; Mcl-1 = Induced myeloid leukemia cell differentiation protein; PDE3A = cGMP-inhibited 3′,5′-cyclic phosphodiesterase A; GPNMB = Transmembrane glycoprotein NMB; FCN1 = Ficolin-1; MAPK14 = Mitogen-activated protein kinase 14; NCAM-L1 = Neural cell adhesion molecule L1; IGFBP-2 = Insulin-like growth factor-binding protein 2. ^2^ The two p-values are for the difference in mean levels and difference in longitudinal trajectories between Pred-treated and Dfz-treated subjects, respectively.

**Table 4 jpm-10-00164-t004:** Summary for biomarker signal confirmation using ELISA.

Protein Name (Uniprot ID)	SOMAscan^®^ Data			ELISA Data			Function
	*p*-Value ^1^: Difference in Mean Levels	*p*-Value ^1^: Difference in Trajectories	Number of Patients/Samples	*p*-Value ^1^: Difference in Mean Levels	*p*-Value ^1^: Difference in Trajectories	Number of Patients/Samples	
MMP-3 (P08254)	0.008	0.216	8 Dfz, 7 Pred/35 samples	0.022	0.378	17 Dfz, 12 Pred/76 Samples	Extracellular matrix degradation
IGFBP-2 (P18065)	0.328	0.04	8 Dfz, 7 Pred/35 samples	0.744	0.0507	10 Dfz, 10 Pred/49 Samples	Regulates growth
IGF-I (P05019)	0.007	0.096	8 Dfz, 7 Pred/35 samples	0.246	0.137	17 Dfz, 12 Pred/75 Samples	Promotes growth

^1^ Two *p*-values are provided for difference in mean levels and difference in longitudinal trajectory slopes between Pred-treated and Dfz-treated subjects for both SOMAscan^®^ and ELISA^®^ analysis, respectively.

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
