# Peer review of "Comparison of Serum Pharmacodynamic Biomarkers in Prednisone-Versus Deflazacort-Treated Duchenne Muscular Dystrophy Boys"

_jpm, 2020, doi:10.3390/jpm10040164_

Round 1

Reviewer 1 Report

JPM 91005 manuscript review:

This manuscript titled as, ‘Comparison of Serum Pharmacodynamic Biomarkers in Prednisone- versus Deflazacort-treated Duchenne Muscular Dystrophy Boys’ by Shefa Tawalbeh et al has novel findings and may deserve consideration for publication.

Title: Good

Abstract: Good

Keys: Good

Introduction: Overall, it is good.

1.Rewrite the following sentence (there are many ‘and’):

Figure 1 shows structures of prednisone and deflazacort and their respective active metabolites prednisolone and 21-desacetyl deflazacort

  1. Add a ‘comma’ between ‘height weight’ in the following sentence:

These studies compared clinical outcome measures like changes in height weight, Cushingoid features, and erythema, among others to evaluate differences in safety

  1. Mini table to list the name of the study, treatment, clinical parameters and treatment outcome can add strength to this timely manuscript.

Methods: Overall, it is good.

1.. The following sentence does not need to be in parentheses

(six-minute walk distance and NorthStar Ambulatory Assessment score were measured for a small subset of patients enrolled later during the study)

  1. If possible, the authors may want to present the variation in dose and schedules of Pred or Dfz treatment and discuss it in the discussion as well.
  2. The following approach may need to be discussed in the discussion, as it may interfere in the overall interpretation of the data, as switching regiments may change end organ response and biomarker expression:

‘For consistency, if a subject switched regimens (e.g., Pred to Dfz), their samples/measurements from prior to the switch were used’.

Results: overall, it is good.

It will be useful, if the authors can provide fold changes for the differentially altered (levels of upregulation or downregulation) biomarkers to appreciate the biomarker evaluation for patient care.

Longitudinal trajectory of serum PD biomarkers in prednisone vs. deflazacort treated DMD boys: Well presented.

Effect of Dfz and Pred on height and weight of DMD boys: Good

Data validation using ELISA: Good

For those biomarkers that were not validated by ELISA, brief literature validation (table showing studies that mention about these biomarkers with reference to clinical parameter (like height, weight and bone density etc) may be useful.

Table 1: Good

Table 2: Good

Table 3: Good

Table 4: Good

Figure 1: Good

Figure 2: Good

Figure 3: Good

Figure 4: Good

Figure 5: Good

Discussion: Overall, it is good.

If possible, the authors may add brief schematic flow chart about possible pathways altered (as indicated by altered biomarkers) during the Pred and Dfz treatments.

Conclusions: Good

Abbreviations for various biomarkers need to be expanded at some place in the manuscript as allowed or required by the journal. Some are shown in the results section. The rest can be shown at the end of conclusion as a separate segment of the manuscript.

Reviewer 2 Report

Tawalbeh, ed al. Try to compare effects  of daflazacort and prednisone on patients affected by DMD. They found that daflazacort is more effective and a bit less toxic than prednisone. They identified some biomarkers which are differentially expressed between two treatments at different times. Authors focused only on three out of seventeen biomarkers:

IGFBP-2; MMP-3; IGF-I which are altered Also in other diseases such as diabetes.

Authors write that a possible limit of this study is the limited cohort of patients analysed.  

 I Think that manuscript could be a masterpiece in the field of glucocorticoids treatment for DMD. However, authors focused only on DMD patients. And the control healthy donors? What is their response to the treatment? Are the same biomarkers altered Also in healthy donors treated with daflazacort and prednisone? To improve scientific soundness of this manuscript I suggest:

  1. Extensive editing of English language  as there  are many typos expecially in introduction and discussion.
  2. Analysis of other biomarkers expression (3 out of 17 are too less).
  3. The analysis must be performed on both patients and healthy controls. Use at least 20/25 of each at 4-10 years of age to check all biomarkers. 

Reviewer 3 Report

In the current study, authors have evaluated the pharmacodynamic markers of Prednisone (Pred) and Deflazacort (Dfz)  in DMD patients. They identified 17 additional biomarkers along with other known markers. The study is important in regard to the management of therapeutic toxicity and response.

I have the following minor concerns:

1-Results needs to be presented in bar-graphs with proper statistics.

2-Authors should discuss the physiological importance of newly identified markers, are they just bystander or have some functional importance.

3-Figure legends need to be more descriptive.

4- The introduction section needs to be more focused.

Round 2

Reviewer 2 Report

Authors replied partially to my conceptual doubts about the impossibility to use healthy donors and the low number of samples analyzed.

However, I expect that authors will amplify their analysis as soon as possible to confirm all the data on a higher and more statistically significant number of samples. This will improve the scientific soundness of your protocols and could be probably useful to discover also other biomarkers in different patients.

The manuscript is now more understandable after English language revision.
